# Identification of Immunogenic Linear B-Cell Epitopes in *C. burnetii* Outer Membrane Proteins Using Immunoinformatics Approaches Reveals Potential Targets of Persistent Infections

**DOI:** 10.3390/pathogens10101250

**Published:** 2021-09-28

**Authors:** Sílvia da Silva Fontes, Fernanda de Moraes Maia, Laura Santa’Anna Ataides, Fernando Paiva Conte, Josué da Costa Lima-Junior, Tatiana Rozental, Matheus Ribeiro da Silva Assis, Adonai Alvino Pessoa Júnior, Jorlan Fernandes, Elba Regina Sampaio de Lemos, Rodrigo Nunes Rodrigues-da-Silva

**Affiliations:** 1Laboratory of Monoclonal Antibodies Technology, Immunobiological Technology Institute, FIOCRUZ, Rio de Janeiro 21040-900, Brazil; silviasilvafontes@gmail.com (S.d.S.F.); fmmaia15@gmail.com (F.d.M.M.); santannalaura@gmail.com (L.S.A.); 2Pilot Plant Implantation Project, Immunobiological Technology Institute, FIOCRUZ, Rio de Janeiro 21040-900, Brazil; fernando.conte@bio.fiocruz.br; 3Laboratory of Immunoparasitology, Oswaldo Cruz Institute, FIOCRUZ, Rio de Janeiro 21040-900, Brazil; josue@ioc.fiocruz.br; 4Laboratory of Hantaviroses and Rickettsioses, Oswaldo Cruz Institute, FIOCRUZ, Rio de Janeiro 21040-900, Brazil; rozental@ioc.fiocruz.br (T.R.); matheus.assis@ioc.fiocruz.br (M.R.d.S.A.); adonai@ioc.fiocruz.br (A.A.P.J.); jorlan@ioc.fiocruz.br (J.F.); elemos@ioc.fiocruz.br (E.R.S.d.L.)

**Keywords:** Q fever, immunoinformatics, B-cell epitope, serodiagnosis, synthetic peptides

## Abstract

*Coxiella burnetii* is a global, highly infectious intracellular bacterium, able to infect a wide range of hosts and to persist for months in the environment. It is the etiological agent of Q fever—a zoonosis of global priority. Currently, there are no national surveillance data on *C. burnetii*’s seroprevalence for any South American country, reinforcing the necessity of developing novel and inexpensive serological tools to monitor the prevalence of infections among humans and animals—especially cattle, goats, and sheep. In this study, we used immunoinformatics and computational biology tools to predict specific linear B-cell epitopes in three *C. burnetii* outer membrane proteins: OMP-H (CBU_0612), Com-1 (CBU_1910), and OMP-P1 (CBU_0311). Furthermore, predicted epitopes were tested by ELISA, as synthetic peptides, against samples of patients reactive to *C. burnetii* in indirect immunofluorescence assay, in order to evaluate their natural immunogenicity. In this way, two linear B-cell epitopes were identified in each studied protein (OMP-H_(51–59)_, OMP-H_(91–106)_, Com-1_(57–76)_, Com-1_(191–206)_, OMP-P1_(197–209)_, and OMP-P1_(215–227)_); all of them were confirmed as naturally immunogenic by the presence of specific antibodies in 77% of studied patients against at least one of the identified epitopes. Remarkably, a higher frequency of endocarditis cases was observed among patients who presented an intense humoral response to OMP-H and Com-1 epitopes. These data confirm that immunoinformatics applied to the identification of specific B-cell epitopes can be an effective strategy to improve and accelerate the development of surveillance tools against neglected diseases.

## 1. Introduction

*Coxiella burnetii* is a polymorphic and obligate intracellular Gram-negative bacterium that is highly infectious and can persist in the environment for months. It is the causative agent of Q fever in humans and coxiellosis in animals—a worldwide disease that is 1 of 13 global priority zoonoses [1,2,3]. In addition, *C. burnetii* is classified as a potential agent of bioterrorism, due to its remarkable resistance to environmental stress, extremely low infectious dose, and ease of dissemination [4,5,6]. A wide range of animals is known to harbor *C. burnetii*, including many species of ticks, birds, and wild, domestic, and companion mammals [7,8]. Infected animals excrete bacteria in large quantities in their birth products, milk, urine, and feces, contaminating the environment. The main transmission route of *C. burnetii* to humans is through inhalation of aerosols contaminated by amniotic fluid and animal placenta, but other possible routes are under investigation, including consumption of raw milk and unpasteurized dairy products, blood transfusions, nosocomial transmission, during autopsies, and delivery of infected pregnant women [9,10,11,12].

Q fever is an ubiquitous zoonosis, with outbreaks reported all over the globe, except for New Zealand and Antarctica [1]. In animals, coxiellosis is often asymptomatic in livestock, but is associated with reproductive disorders causing miscarriages, premature births, infertility, mastitis, and decreased milk production, especially in ungulates and ruminants [13]. In humans, clinical outcomes vary in severity, ranging from asymptomatic infection with seroconversion, to acute Q fever—a mild, influenza-like, self-limited, febrile illness that can progress to more severe cases, atypical pneumonia, or hepatitis [14]. In both asymptomatic and acute cases, Q fever can progress as a persistent and localized infection of a specific organ—mainly infectious endocarditis—and vascular infection, although other impairments have been described that may result in death, depending on host characteristics [1,15]. To date, since the 2007 epidemic in the Netherlands [16], the number of global publications on Q fever has increased. However, despite having been listed as a notifiable disease by the World Organization for Animal Health [17], in both animals and humans, *C. burnetii* infections remain poorly understood, and their prevalence still is underestimated in several regions of the world [18]. Moreover, most of the countries where Q fever exists are yet to formulate sanitary control measures to control the disease in livestock and humans.

Although most human cases are asymptomatic, Q fever may manifest nonspecific symptoms that could lead to misdiagnosis with other prevalent tropical diseases, such as dengue, malaria, or leptospirosis [19]. In Brazil, Q fever is a mandatory notification disease, in the context of differential diagnosis with Brazilian spotted fever; however, there is still little information about *C. burnetii*’s circulation in Brazil. In recent years, case reports and seroprevalence studies reported the detection of *C. burnetii* in animals [20,21,22,23], humans [23,24,25,26,27] and, most recently, in artisanal cheese [28] and unpasteurized milk [29]. These data, allied to the low number of Q fever cases reported annually, suggest that Q fever might be underreported, and reinforce the urgency of the development of novel diagnostic tools that allow for large-scale epidemiological surveillance, as well as improving the inspection of products of animal origin.

Regarding the diagnosis of Q fever, the isolation of *C. burnetii* is rarely attempted due to its prolonged incubation period and the biosafety level required [30,31]. Currently, the Q fever/coxiellosis diagnosis is mainly based on serological (immunofluorescence assay, ELISA, and latex agglutination assay) and molecular methods (PCR). However, the poor field applicability, along with the inability to differentiate active infection from recovered individuals, or even acute and chronic Q fever, pose serious bottlenecks to the large-scale surveillance of Q fever/coxiellosis [32,33,34,35]. In this context, numerous novel antigen candidates have been proposed to improve the serodiagnosis of Q fever [36,37,38,39,40]. Among these antigens, OMP-H (CBU_0612) was described as an immunodominant marker for acute and persistent forms of Q fever [35,41], Com-1 (CBU_1910) is considered a reliable Q fever serodiagnosis marker [42,43], and OMP-P1 (CBU_0311) is a porin—highly expressed only in the replicative form of the bacterium in the cell hosts [44,45,46]. The identification of B-cell epitopes arises as a promising alternative to improve the specificity of serological tests to *C. burnetii*, since the use in serodiagnosis of the above-mentioned whole antigens may result in cross-reactivity with other phylogenetically related proteobacteria.

Recently, synthetic peptides have emerged as novel targets for the efficient serological diagnosis of infectious diseases of viral, bacterial, and parasitic origin [47,48,49,50,51,52,53,54]. This kind of molecule is applied via latex agglutination assay to diagnose Q fever, presenting good sensitivity and specificity for diagnosis in cattle [49,55]. This approach, compared to diagnosis based on whole antigens, presents low-cost production, higher specificity and reproducibility with no batch-to-batch variation, and large-scale production [56,57,58]; however, it still depends on the accurate identification of more immunodominant epitopes. Therefore, this study aimed to identify B-cell epitopes from *C. burnetii* outer membrane proteins (OMP-P1, OMP-H, and Com-1) *in silico*, and to validate them against samples from Brazilian exposed individuals.

## 2. Results

### 2.1. Prediction of Linear B-Cell Epitopes in C. burnetii OMPs

In order to improve the accuracy of *in silico* identification of epitopes, we used a combination of three prediction algorithms: BepiPred 1.0, ABCpred, and ESA. Firstly, 10 sequences were predicted as linear B-cell epitopes in studied proteins—2 sequences in OMP-H, and 4 sequences each in Com-1 and OMP-P1. Among all 10 sequences, 4 were predicted by all algorithms, while 3 were predicted by Bepipred and ABCpred, 2 by ABCpred and ESA, and 1 potential epitope by Bepipred and ESA (Table 1).

### 2.2. Prediction of Epitopes’ Antigenicity and Specificity

The antigenicity of predicted linear epitopes was evaluated using the VaxiJen algorithm. Based on this evaluation, eight predicted epitopes were considered antigenic, among which were the four sequences predicted as linear B-cell epitopes by the three used algorithms (OMP-H_(51–59)_, OMP-H_(98–106)_, Com-1_(57–76)_, and Com-1_(191–206)_). The sequences Com-1_(83–96)_ and OMP-P1_(43–51)_ presented VaxiJen scores lower than the threshold (0.139 and 0.383, respectively), and were considered non-antigenic and excluded from the study (Table 2).

Additionally, in order to exclude epitopes conserved among proteobacteria, predicted epitopes were evaluated for their degree of conservation using BLASTp, considering a cutoff *E*-value of 1.0. All sequences predicted as antigenic linear B-cell epitopes were specific to *C. burnetii*, since no significant similarities were found when compared to phylogenetically similar bacteria and microbiota bacteria.

### 2.3. Epitope Exposition in Protein Quaternary Structure

Considering that OMPs naturally oligomerize in order to exclude sequences that interact with other chains of each oligomer from the study, we evaluated the localization of the predicted epitopes in protein quaternary structures. Firstly, oligomerization analysis indicates that each of the studied proteins—OMP-H, OMP-P1, and Com-1—forms homotrimers. Here, aiming to selected immunodominant epitopes, and considering contacts between chains within 3.0 A, sequences that presented more than 30% of their amino acids interacting with other chains were excluded from the study.

Regarding the exposition of epitopes on protein quaternary structures, predicted epitopes OMP-H_(51–59)_ and OMP-H_(91–106)_ (Figure 1a,b), Com-1_(191–206)_ (Figure 1c,d), and OMP-P1_(197–209)_ and OMP-P1_(215–227)_ (Figure 1e,f) were exposed on the trimer surface, and did not interact with other chains in the oligomeric structure; meanwhile, the epitopes Com-1_(57–76)_, Com-1_(26–34)_ (Figure 1d), and OMP-P1_(98–106)_ (Figure 1e) presented, 25%, 78%, and 56%, respectively, of their sequences interacting with other chains when the predicted OMP trimeric structure was analyzed. Therefore, predicted epitopes Com-1_(26–34)_ (Figure 1d) and OMP-P1_(98–106)_ (Figure 1f) were considered buried within the quaternary structures, and were excluded from the study.

### 2.4. Studied Population Description

The studied population was composed of 57 suspected Q fever cases; among them, 26 individuals (45.6%) were *C. burnetii* seroreactive in IFA, with antibody titers ranging from 1:64 to 1:32,768 (median: 1:128), while 31 patients (54.4%) were non-reactive to *C. burnetii*, and were used as the negative control group. As shown in Table 3, both groups—reactive and non-reactive to *C. burnetii*—presented a similar median age (33 years old).

Regarding clinical manifestations, fever and prostration were reported by 10 (38%) and 6 (23%) Q fever cases, respectively, and by 14 (45%) and 12 (39%) non-reactive individuals, respectively (Table 3). Moreover, endocarditis was diagnosed in 4 (15%) *C. burnetii*-reactive patients and in 2 (6%) non-reactive individuals. 

### 2.5. Preliminary Assessment of the Potential of Epitopes as Serological Antigens

Aiming to verify the natural immunogenicity of the predicted epitopes, samples of patients both reactive and non-reactive to *C. burnetii* were tested by ELISA against synthetic peptides containing predicted linear B-cell epitopes. Remarkably, based on our threshold, we observed no responsivity against peptides in the negative control group, composed of *C. burnetii*-non-reactive individuals (Figure 2a). Moreover, in the *C. burnetii*-reactive group, the frequencies of response against the epitopes OMP-H_(51–59)_ (65%), OMP-H_(91–106)_ (50%), Com-1_(57–76)_ (58%), Com-1_(191–206)_ (58%), and OMP-P1_(215–227)_ (15 responders; 58%) were higher than 50%. Moreover, the frequency of response to OMP-P1_(197–209)_ (23%) was statistically lower than the frequencies of response to OMP-H_(51–59)_ (*p* = 0.0047), Com-1_(57–76)_, Com-1_(191–206)_, and OMP-P1_(215–227)_ (*p* = 0.0227) (Figure 2b). Despite these differences, we observed a similar magnitude of IgG-RI against synthetic peptides, ranging from 1.01 to 6.4, in which the median reactivity indices were 1.37 (IQ: 1.095–2.055) against OMP-H_(51–59)_, 1.74 (IQ: 1.37–2.045) against OMP-H_(91–106)_, 1.4 (IQ: 1.14–1.95) against Com-1_(57–76)_, 1.68 (IQ: 1.18–2.26) against Com-1_(191–206)_, 1.39 (IQ:1.31–1.46) against OMP-P1_(197–209)_, and 1.69 (IQ: 1.22–2.01) against OMP-P1_(215–227)_ (Figure 2c).

As shown in Table 4, 77% of studied individuals (*n* = 20) presented antibodies against at least one of the identified epitopes. Regarding the reactivity against each studied protein epitope, 18 patients (69%) responded to at least one of the OMP-H epitopes, 17 patients were reactive to at least one of the Com-1 epitopes, while 15 individuals presented antibodies against at least one of the OMP-P1 epitopes.

### 2.6. Associations between Humoral Response to Synthetic Peptides and Clinical Features

As previously shown in Table 4, between 20% and 33% of responders to the epitopes OMP-H_(51–59)_, OMP-H_(91–106)_, Com-1_(57–76)_, Com-1_(191–206)_, and OMP-P1_(215–227)_ presented high levels of specific antibodies. Aiming to explore these data, we compared the clinical features of high responders (HR: RI > 2), low responders (LR: 1 > RI ≤ 2), and non-responders (NR: RI ≤ 1) to each studied peptide. Remarkably, 75% of patients reactive to *C. burnetii* who presented endocarditis (*n* = 6) also demonstrated a high level of antibodies to at least one of the identified epitopes. The frequency of endocarditis seems to be higher in HR than in LR and NR to OMP-H_(51–59)_ (HR: 40%; LR: 8% and NR: 11%; Figure 3a), OMP-H_(91–106)_ (HR: 50%, LR: 11% and NR: 8%; Figure 3b), Com-1_(57–76)_ (HR: 67%, LR: 8% and NR: 9%; Figure 3c), and Com-1_(191–206)_ (HR: 40%, LR: 10% and NR: 9%; Figure 3d). Moreover, despite the six responders to OMP-P1_(197–209)_ having presented a low specific antibody level, 50% presented endocarditis, while only 5% of NR to this epitope presented this complication (Figure 3e).

Regarding another clinical feature, prostration seems to be more frequent in patients who present antibodies against OMP-H_(51–59)_ (HR: 40%, LR: 33%; NR: 0; Figure 3a), OMP-H_(91–106)_ (HR: 50%, LR: 33% and NR: 8; Figure 3b), and Com-1_(57–76)_ (HR:33%, LR:33% and NR: 9%; Figure 3c). Remarkably, an apparently minor hospitalization rate was observed in high responders to Com-1_(57–76)_ (HR:0, LR: 50%, NR: 36%; Figure 3c) and Com-1_(191–206)_ (HR: 20%, LR: 50%, NR: 36%; Figure 3d). Finally, the antibody response against OMP-P1_(215–227)_ seems not to be related to clinical features, with no differences between HR, LR, and NR to this epitope (Figure 3f).

## 3. Discussion

Most *C. burnetii* infections are asymptomatic or associated with nonspecific symptoms. Based on this, seroprevalence surveys arise as a potential way to evaluate the real prevalence of *C. burnetii* [19]. However, despite studies in the Netherlands [59,60], Australia [61], the USA [62], Bhutan [63], Northern Ireland [64], Cyprus [65], and South America [66,67], the true global prevalence of *C. burnetii* infections is still unknown due to the dearth of representative national seroprevalence surveys in the world. In this scenario, the identification of linear B-cell epitopes arises as a potential approach to improve and accelerate the development of novel diagnostic tests that could allow effective *C. burnetii* seroprevalence surveys.

In this study, we identified B-cell epitopes in antigen candidates for the development of serological tests for *C. burnetii*, and showed their immunogenicity in individuals naturally exposed to infection. In this way, we explored three *C. burnetii* outer membrane proteins (OMPs): OMP-H (CBU_0612), an outer membrane chaperon protein highlighted as a very strong immunodominant marker for both acute and persistent forms of Q fever [35,41]; Com-1 (CBU_1910), a strongly immunogenic protein [38,40] that is considered a reliable Q fever serodiagnosis marker [42,43]; and OMP-P1, a porin, highly expressed in large-cell variants (LCVs)—the replicative form of the bacterium in the cell hosts—and downregulated in small-cell variants (SCVs), the metabolically inactive and resistant form found in the environment [44,45,46]. However, the use of whole antigens in serodiagnosis may result in cross-reactivity with other related proteobacteria. Based on this, the B-cell epitopes’ identification arises as a promising alternative to improve the specificity of serological tests for *C. burnetii*.

In recent years, combinations of prediction algorithms have been used to improve the accuracy of linear B-cell epitope prediction against viruses [68,69,70,71], fungi [72], protozoa [73,74], and bacteria [75,76,77]. However, there was only one *in silico* study of *C. burnetii*, in which six different algorithms were individually used to explore OMP-H and Com-1, predicting between 1–7 linear B-cell epitopes in each studied protein [78]. In this study, using a combination of prediction algorithms—Bepipred, ABCpred, and Emini Surface Accessibility prediction—four sequences (OMP-H_(51–59)_, OMP-H_(91–106)_, Com-1_(57–76)_, and Com-1_(191–206)_) were predicted as epitopes by the three used algorithms, while six sequences were predicted by only two of them. Moreover, combining this initial epitope identification with the antigenic analysis, we predicted only two epitopes in OMPH, three in Com-1, and three in OMP-P1. This reduction in the number of predicted epitopes in comparison to those reported by Jaydari et al. [78] may be related to the improvement in the accuracy of *in silico* analyses by the combination of prediction algorithms. Moreover, the specificities of the predicted epitopes were verified by comparison with proteobacteria proteins. Remarkably, despite all of the predicted epitopes being conserved among *C. burnetii* strains described in the UniProt database (data not shown), the scarcity of studies concerning OMPs’ polymorphism hampers conclusions about the real conservation of the identified epitopes in *C. burnetii* strains.

Furthermore, in a previous study, our group hypothesized that predicted epitopes that fail in experimental validation could be buried in protein quaternary structures [71], upon which prediction algorithms would be unable to evaluate these structures. Based on this hypothesis, and considering the oligomerization of similar proteins [45,79,80], we evaluated the localization of predicted epitopes in oligomeric structures of studied proteins. To do this, we modeled the quaternary structures of the studied proteins, and observed that epitopes Com-1_(26–34)_ and OMP-P1_(98–106)_ were buried in Com-1 and OMP-P1 homoligomers (Figure 1). These two epitopes—Com-1_(26–34)_ and OMP-P1_(98–106)_—were excluded from the study, while the other six predicted epitopes (OMP-H_(51–59)_, OMP-H_(91–106)_, Com-1_(57–76)_, Com-1_(191–206),_ OMP-P1_(197–209)_, and OMP-P1_(215–227)_) were selected and experimentally validated. From our point of view, the evaluation of the predicted epitopes’ exposition in the protein oligomeric structure can be a critical step to improve the accuracy of epitope prediction, and the lack of those analyses could explain—at least in part—the low validation rates (15%) of predicted epitopes for some infectious agents seen in *Brucella* sp. proteins [81,82,83]. On the other hand, despite the improvements in methodologies and the results of *in silico* studies, the evaluation of reactivity to predicted linear B-cell epitopes remains rare, and this was the first study using samples from Brazilian *C. burnetii*-exposed individuals to confirm the natural immunogenicity of B-cell epitopes predicted in this bacterium’s proteins.

In Brazil, although the presence of *C. burnetii* has been demonstrated in animals [20,21,22,23], humans [21,23,24,25,26,27] and, most recently, in artisanal cheese [28] and unpasteurized milk [29], Q fever remains poorly understood and reported. Moreover, despite the possibility of mistaking Q fever for influenza, dengue, malaria, leptospirosis, and other infectious diseases [30], in Brazil, only suspected rickettsiosis cases are investigated as Q fever, corroborating the underreporting of this zoonosis, and explaining the limited number of *C. burnetii*-reactive samples in this study (*n* = 26). Remarkably, similar-sized samples were previously used to successfully define *C. burnetii* proteins as antigens to serological tests [37,84,85,86], indicating that the number of patients used in our study was sufficient to prove the natural immunogenicity of the identified epitopes.

In this way, samples of the studied population were tested by ELISA against synthetic peptides, representing the sequences of six predicted linear B-cell epitopes (OMP-H_(51–59)_, OMP-H_(91–106)_, Com-1_(57–76)_, Com-1_(191–206),_ OMP-P1_(197–209)_, and OMP-P1_(215–227)_). Our data showed high specificity of the selected B-cell epitopes, such that all unreactive patients to C. *burnetii* according to IFA were also non-responsive against the tested synthetic peptides (Figure 2a). Moreover, all of the synthetic peptides were recognized by *C. burnetii*-reactive patients, confirming their natural immunogenicity. In this context, a previous study using a synthetic peptide containing the sequence of epitope Com-1_(57–76)_ in a latex agglutination test (LAT) reported a performance (sensitivity and sensibility) of ~75% in cattle samples, when compared to a commercial ELISA test [49]. Although more studies are necessary to evaluate the true specificity and sensitivity of these epitopes, immunodominant epitope combinations can be used to improve the accuracy of serological diagnosis of *C. burnetii*. This strategy can be applied especially to commercial ELISA tests for *C. burnetii*, which have presented low sensitivity and specificity when compared to IFA [87,88]. Here, while the reactivity to a single peptide ranged from 23% to 65% of studied patients, the combination of epitopes increased sensitivity, allowing the detection of specific antibodies against at least one epitope of the same protein in ~60–70% of *C. burnetii*-seroreactive individuals. Moreover, 77% of the *C. burnetii*-reactive group was also reactive to at least one of the identified epitopes (Table 4). These data reinforce the need to identify linear B-cell epitopes in other promising serological markers—such as OmpA [89], *C. burnetii* macrophage infectivity potentiator protein (Cb-Mip) [90], and YbgF [85]—in order to improve the development of novel tools for the diagnosis of *C. burnetii*.

Despite the similarities in the magnitude of response against the identified epitopes, where the epitopes OMP-H_(51–59)_, OMP-H_(91–106)_, Com-1_(57–76)_, Com-1_(191–206)_, and OMP-P1_(215–227)_ were recognized in more than 50% of the studied samples, we observed that ~20–30% of responders presented a high level of antibodies (R.I. > 2.0), while all responders to OMP-P1_(197–209)_—the peptide with lower recognition rates (Figure 2b)—presented a low level (1.0 > RI ≤ 2.0) of specific antibodies against this peptide. The frequency of high responders to specific antigens is not well explored for *C. burnetii*; however, grouping individuals according to RI was a strategy previously employed by our group in a study that reported high responders against *Plasmodium vivax* recombinant protein, suggesting an association between frequency of recently exposed individuals and high humoral response to this protein [91]. Based on this previous experience, in order to investigate associations between clinical features and the level of antibodies against the identified epitopes, we evaluated the frequency of cases of endocarditis, fever, prostration, and hospitalization among patients reactive to *C. burnetii*, comparing these features between individuals who presented high levels of antibodies (R.I. > 2), low levels of antibodies (1.0 > R.I. ≤ 2.0), and who were non-responsive (R.I. ≤ 1) to an identified epitope. Remarkably, we observed that 75% of the studied patients with endocarditis presented a high level of antibodies against at least one of the identified epitopes, corroborating that the intense humoral response may be a marker in chronic/persistent cases of Q fever, which was already suggested by the antibody response against Com-1 recombinant protein [43].

Regarding the response against each epitope, we observed that endocarditis seems to be more frequent in patients who are HR to the OMP-H and Com-1 epitopes (Figure 3), suggesting that an intense humoral response to the Com-1 and OMP-H epitopes may be associated with chronic Q fever complications. This hypothesis corroborates the potential of recombinant OMP-H as an antigen for the serological diagnosis of *C. burnetii* endocarditis cases [37]; however, more studies involving a higher number of patients and complications are still necessary to prove this hypothesis. Moreover, we observed low hospitalization rates in high responders to Com-1 epitopes, while we observed a low frequency of prostration reports in non-responders to OMP-H epitopes when compared to LR and HR to OMP-H epitopes. Finally, the humoral response against OMP-P1 seems not to be associated with clinical features, given that OMP-P1_(197–209)_ was the epitope with the lowest recognition rate, and did not present high responders, while we observed no differences in clinical features among HR, LR, and NR to OMP-P1_(215–227)_. These data corroborate the study of Qingfeng Li et al., which showed that recombinant OMP-P1 induced a low antibody level, but a high cellular immune response, in immunized mice [92]. Here, we observed that OMP-P1 linear B-cell epitopes are poorly immunogenic, but more studies aiming to identify its TCD4 and TCD8 epitopes are still necessary, and may be used to improve vaccine development [93].

In conclusion, despite the limitations in sample size and study design, six linear B-cell epitopes were identified by a combination of in silico tools and, subsequently, confirmed as naturally immunogenic in individuals exposed to *C. burnetii*. This was the first study evaluating the natural immunogenicity of the predicted linear B-cell epitopes in the *C. burnetii* outer membrane proteins OMP-H, Com-1, and OMP-P1 using samples of human patients. Therefore, more serosurveys regarding these epitopes are encouraged, and are essential in order to estimate their real specificity and sensitivity. Currently, synthetic peptides representing B-cell epitope sequences are considered a valuable tool as novel molecules for the reliable and rapid diagnosis of infectious diseases [58,94], with promising results for Q fever/coxiellosis [49,55], as corroborated by this study. When compared to whole proteins, the use of synthetic peptides as diagnostic antigens allows a higher specificity and reproducibility with no batch-to-batch variation, along with easy and low-cost production. In this context, we believe that the identification of B-cell epitopes may be an effective strategy to improve and accelerate the development of surveillance tools for Q fever and other neglected diseases.

## 4. Materials and Methods

### 4.1. Sequence Data

For in silico analyzes and 3-dimensional (3D) structure modeling, the full sequences of OMP-H (ID: CBU_0612), Com-1 (CBU_1910), and OMP-P1 (ID: CBU_0311) were obtained from the UniProt database (https://www.uniprot.org/ accessed on 1 June 2021).

### 4.2. B-Cell Epitope Prediction 

Firstly, based on information from the UniProt database, transmembrane regions, signal peptides, and cytoplasmic regions of the studied proteins were excluded from B-cell epitope predictions. The identification of linear B-cell epitopes in the extracellular regions of the studied proteins was performed based on the combination of three different algorithms: Bepipred 1.0 (threshold: 0.35), Emini Surface Accessibility prediction (ESA) (threshold 1.0), and ABCpred (threshold 0.75). Sequences predicted by at least two of the used algorithms were considered predicted B-cell epitopes, and were evaluated for antigenicity using VaxiJen (http://www.ddg-pharmfac.net/vaxijen/VaxiJen/VaxiJen.html accessed on 1 June 2021) (threshold 0.4).

Briefly, Bepipred takes a single sequence in FASTA format input, and each amino acid receives a prediction score based on hidden Markov model profiles of known antigens and incorporates propensity scale methods based on hydrophilicity and secondary structure prediction. For each input sequence, the server outputs a prediction score [95]. Meanwhile, ESA calculates the surface accessibility of hexapeptides, with values greater than 1.0 indicating an increased probability of being found on the surface [96]. Moreover, ABCpred is the first server developed based on a recurrent neural network (machine-based technique), using fixed-length patterns to predict B-cell epitopes in an antigen sequence, with 65.93% accuracy [97]. VaxiJen is the first server for alignment-independent prediction of protective antigens; it was developed to allow classification of antigens based solely on the physicochemical properties of proteins, without recourse to sequence alignment. A bacterial protein dataset, threshold 0.4, was used to derive models for the prediction of whole-protein antigenicity, showing prediction accuracy from 70% to 89% [98,99]. All prediction algorithms were accessed in March 2019.

### 4.3. Evaluation of B-Cell Epitopes’ Degree of Conservation 

To evaluate the degree of conservation of the predicted epitopes, the identified sequences were compared for similarity with bacteria from the phylum Proteobacteria (*Francisella tularensis, Legionella Pneumophila, Escherichia coli, Pseudomonas Aeruginosa, Rickettsia rickettsii, Ehrlichia chaffeensis, Bartonella henselae, Brucella melitensis, Afipia felis,* and *Campylobacter jejuni*) and microbiota bacteria (*Enterobacter* spp., *Klebsiella* spp., *Citrobacter* spp., and *Enterococcus coli*) using BLASTp (https://blast.ncbi.nlm.nih.gov/Blast.cgi?PAGE=Proteins accessed on 21 June 2021). When compared to this group of organisms, the sequences that presented E-values higher than 1 were considered to be non-conserved B-cell epitopes.

### 4.4. Three-Dimensional (3D) Structure Modeling

The 3D structures of the studied OMPs were modeled using the Robetta server (http://new.robetta.org/ accessed on 21 June 2021); this server is continually evaluated through continuous automated model evaluation (CAMEO), and its primary service is to predict the 3D structure of a protein given its amino acid sequence [100,101,102,103]. The PDB formats reported from this server were visualized using PyMOL V1 viewer software [104,105].

### 4.5. Evaluation of Epitopes’ Exposure in the Protein Quaternary Structures

Considering that the OMPs generally form oligomers, the modeled 3D structures of the studied proteins were submitted to the GalaxyHomomer server (http://galaxy.seoklab.org/cgi-bin/submit.cgi?type=HOMOMER accessed on 21 June 2021), which performs comparative modeling of quaternary structures [106,107]. In this study, it was defined that the sequences that had at least 30% of their amino acids interacting with other chains were considered hidden in the quaternary structure of the protein, and were excluded from further analyses. Finally, sequences predicted as antigenic and linear B-cell epitopes—non-conserved among proteobacteria, and exposed in quaternary structures—were selected for experimental validation. The schematic abstract of the prediction of B-cell epitopes is represented in Figure 4.

### 4.6. Peptide Synthesis

Sequences predicted as antigenic linear B-cell epitopes were chemically synthesized by the company GenOne Biotechnologies, Brazil. Analytical chromatography of the peptides demonstrated a purity of approximately > 95%, and mass spectrometry analysis of the peptides indicated their expected mass.

### 4.7. Studied Population

A total of 57 samples of suspected Q fever cases were provided by the Brazilian National Reference Center for Rickettsioses—Laboratory of Hantaviruses and Rickettsiosis—Oswaldo Cruz Foundation. Serum samples were previously tested by IFA for the presence of immunoglobulin G antibodies (IgG) against *Coxiella burnetii*, using a commercial kit from Scimedx^®^ (Dover, NJ, USA) [108]. Firstly, suspected samples were screened at a 1:64 dilution, and those that were considered positive were further diluted to determine the IFA´s end titers. Negative and positive controls were included for each test run. The study was reviewed and approved by the Oswaldo Cruz Foundation Ethical Committee and the National Ethical Committee of Brazil (number CAAE: 39056120.6.0000.5248).

### 4.8. Experimental Confirmation of the Antigenicity of the Predicted Epitopes

Samples of suspected and confirmed Q fever cases were screened for the presence of naturally acquired antibodies against the synthetic peptides via ELISA. Briefly, MaxiSorp 96-well plates (Nunc, Rochester, NY, USA) were coated with 100 μg/mL of a peptide. After overnight incubation at 4 °C, plates were washed with phosphate-buffered saline (PBS) and blocked with PBS-containing 5% non-fat dry milk (PBS-M) for 1 h at 37 °C. Individual plasma samples diluted 1:100 on PBS-M were added in duplicate wells, and the plates were incubated at 37 °C for 2 h. After three washes with PBS-Tween20 (0.05%), bound antibodies were detected with peroxidase-conjugated goat anti-human IgG (Sigma St. Louis, MO, USA), followed by TMB (3,3’,5,5’-tetramethylbenzidine). The absorbance was read at 450 nm using an xMark™ microplate absorbance spectrophotometer (Bio-Rad, Hercules, CA, USA). The results for total IgG were expressed as the reactivity index (RI)—the ratio between the mean optical density (OD) of tested samples and the mean OD of 31 negative controls plus 2 standard deviations (SD). Subjects were considered IgG responders to a particular antigen if the RI was higher than 1.

### 4.9. Statistical Analysis

All statistical analyses were carried out using Prism 5.0 for Windows (GraphPad Software, Inc., San Diego, CA, USA). The one-sample Kolmogorov–Smirnoff test was used to determine whether a variable was normally distributed. Differences in frequencies of IgG responders to synthetic peptides were evaluated using Fisher’s exact test. The Mann–Whitney test was used to compare reactivity indices against synthetic peptides between responders to each epitope. A two-sided *p*-value < 0.05 was considered significant.

## 5. Conclusions

Our data corroborate the use of immunoinformatics approaches to identify epitopes as targets of humoral response that can be further explored in the development of novel surveillance tools for Q fever and other neglected diseases. Here, exploring linear B-cell epitopes in *C. burnetii*—OMP-H, Com-1, and OMP-P1—we experimentally confirmed the natural immunogenicity of two epitopes in each studied protein. Moreover, our data suggest that an intense humoral response to OMP-H and Com-1 epitopes may be related to Q fever complications, such as endocarditis.

## Figures and Tables

**Figure 1 pathogens-10-01250-f001:**
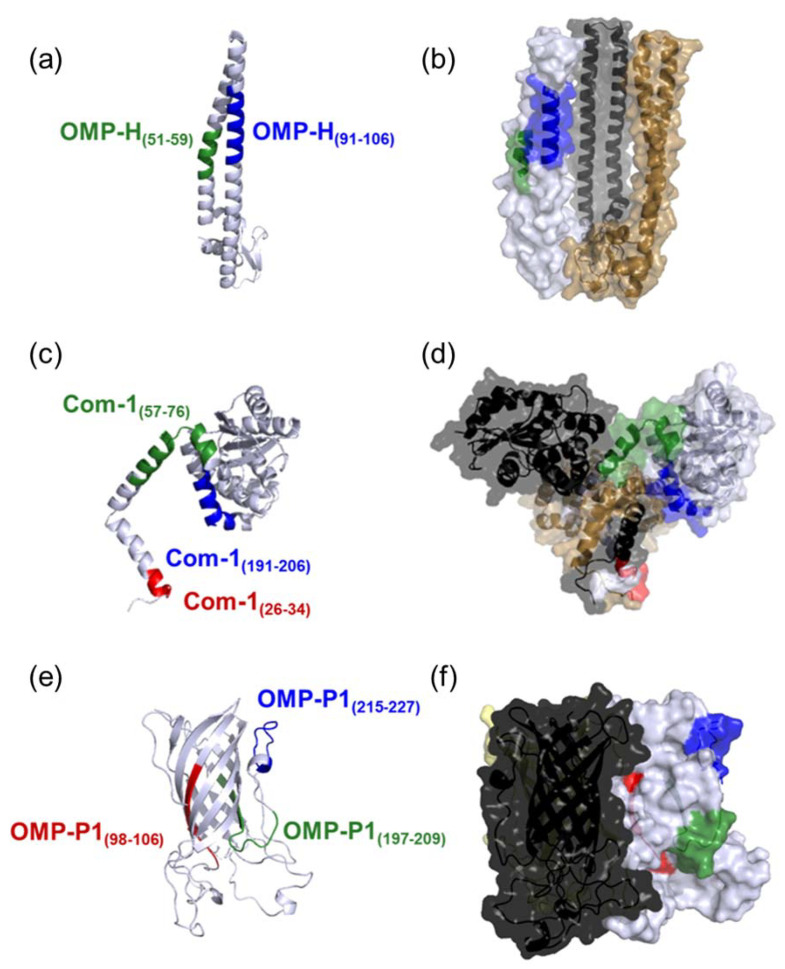
Conformational structure of the studied proteins: (**a**) OMP-H monomeric structure (gray). (**b**) Surface of OMP-H trimeric structure. Epitopes OMP-H_(51–59)_ and OMP-H_(91–106)_ are highlighted in green and blue, respectively. (**c**) Com-1 monomeric structure (gray). (**d**) Surface of Com-1 trimeric structure. Epitopes Com-1_(26–34)_, Com-1_(57–76)_, and Com-1_(191–206)_ are highlighted in red, green, and blue, respectively. (**e**) OMP-P1 monomeric structure (gray). (**f**) Surface of OMP-P1 trimeric structure. Epitopes OMP-P1_(98–106)_, OMP-P1 _(197–209)_, and OMP-P1 _(215–227)_ are highlighted in red, green, and blue, respectively. The chains of trimeric structures (**b**,**d**,**f**) are indicated in gray, gold, and black.

**Figure 2 pathogens-10-01250-f002:**
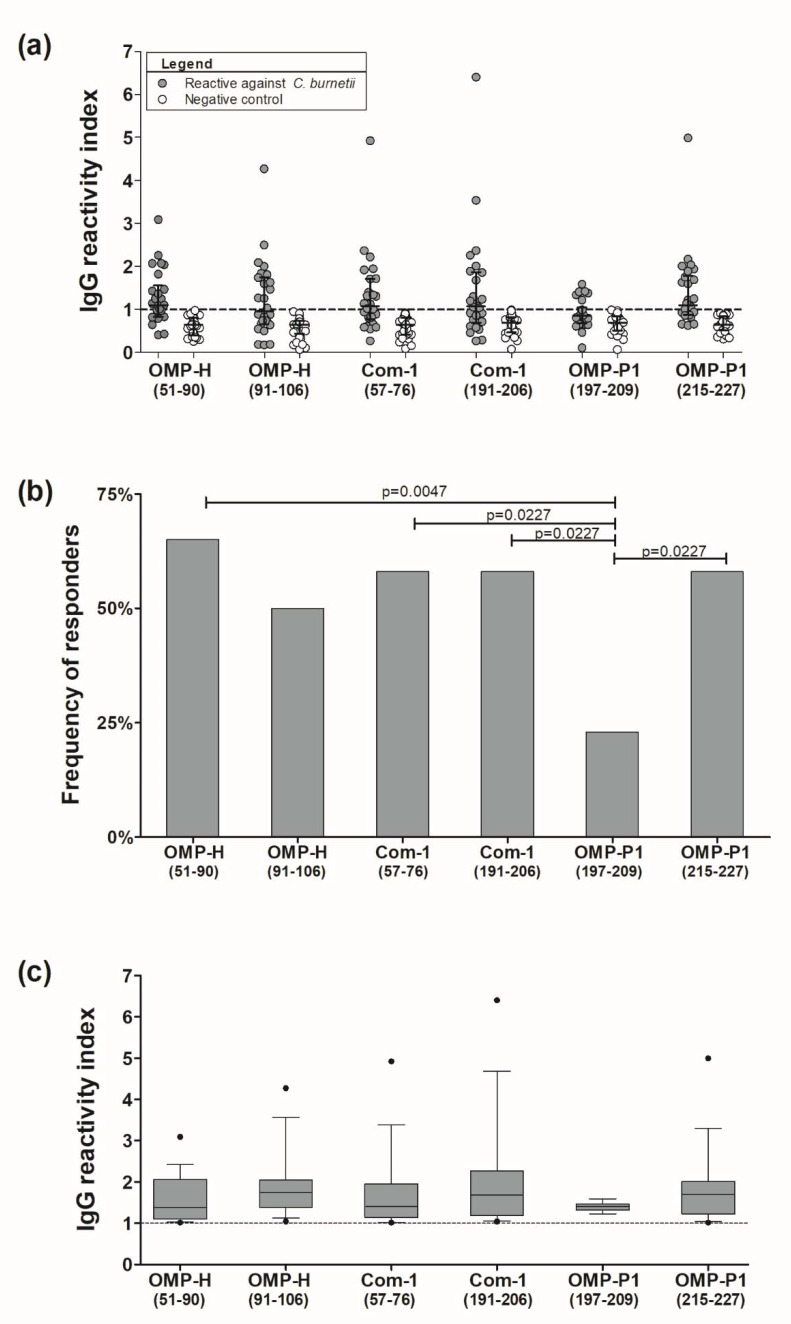
Evaluation of the natural immunogenicity of predicted *C. burnetii* epitopes: (**a**) Reactivity indices against predicted epitopes of patients reactive (gray points) and non-reactive (white points) to *C. burnetii* in IFA. (**b**) Frequencies of response against predicted epitopes. (**c**) The magnitude of IgG response against synthetic peptides. The reactivity indices (RI) represent the ratios between the optical density of each reaction to *C. burnetii* and the cutoff value, defined as the mean of the control group’s optical densities plus twice their standard deviation. Each point represents an individual R.I. against one of the synthetic peptides (OMP-H_(51–59)_, OMP-H_(91–106)_, Com-1_(57–76)_, Com-1_(191–206),_ OMP-P1_(197–209)_,_,_ or OMP-P1_(215–227)_), the traced line indicates the cutoff, while lines represent the median and interquartile range (SEM) of reactive (gray points) and non-reactive (white points) patients to the studied peptides. The comparison of frequencies was done using Fisher´s exact test, and statistical differences were represented by p-values among bars. The magnitudes of response against predicted epitopes are indicated by gray boxes and whiskers (10th–90th percentile), with outliers –indicated by black points in the last graph.

**Figure 3 pathogens-10-01250-f003:**
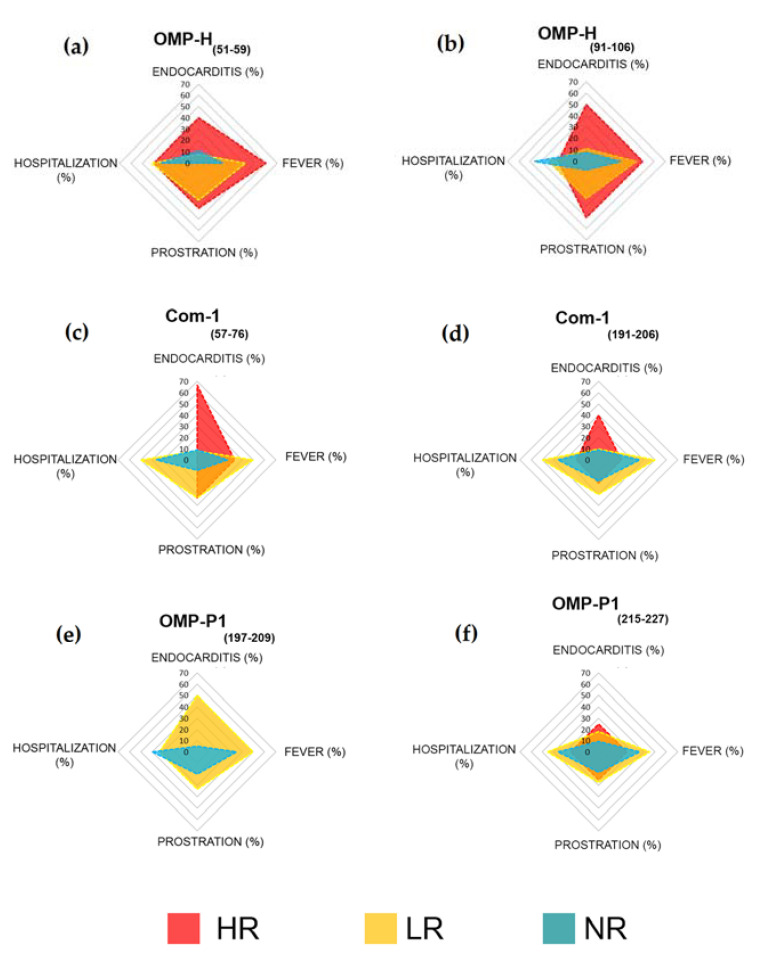
Comparison of age and clinical features between high responders (red), low responders (yellow), and non-responders (blue) to the studied epitopes. Indicated numbers represent the frequencies (%) of reported hospitalizations, fever, prostration, and endocarditis cases in high responders (RI > 2), low responders (1.0 > R.I. ≤ 2.0), and non-responders (R.I. ≤ 1) to the epitopes (**a**) OMP-H_(51–59)_, (**b**) OMP-H_(91–106)_, (**c**) Com-1_(57–76)_, (**d**) Com-1_(191–206)_, (**e**) OMP-P1_(197–209)_, and (**f**) OMP-P1_(215–227)_.

**Figure 4 pathogens-10-01250-f004:**
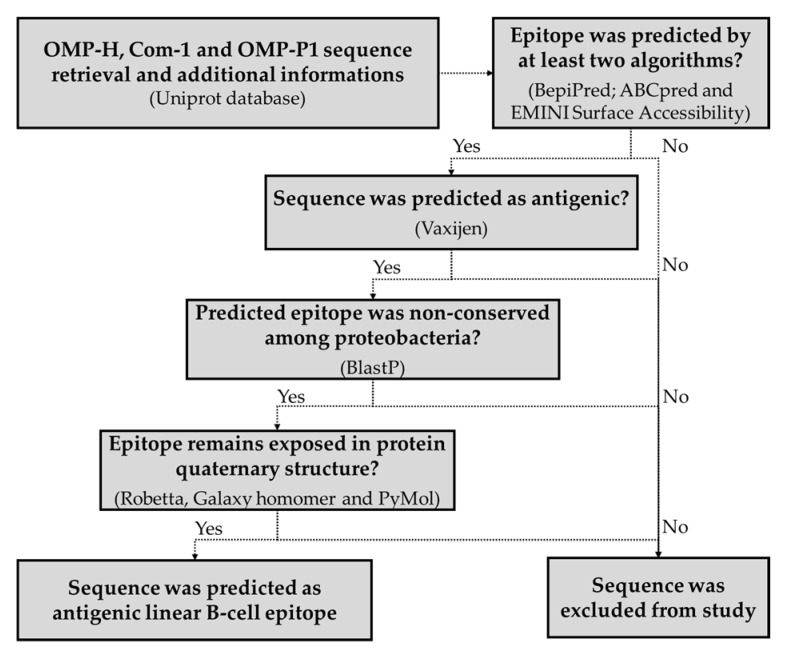
Schematic summary of B-cell epitope prediction.

**Table 1 pathogens-10-01250-t001:** Linear B-cell epitopes identified *in silico* in the *C. burnetii* OMPs.

Protein	Sequence	Bepipred	ABCpred	ESA
OMP-H	51-QFSPQREKM-59	×	×	×
	91-EIQNDESTLRQQQQQF-106	×	×	×
Com-1	26-FSFSPQQVK-34	×	×	-
	57-ALQKKTEAQQEEHAQQAIKE-76	×	×	×
	83-FNDPASPVAGNPHGN-96	×	×	-
	191-KKDMDNPAIQKQLRDN-206	×	×	×
OMP-P1	43-GYKSYTYDQ-51	×	-	×
	98-KAQYQYDNV-106	-	×	×
	197-LSYDYALYRSKSN-209		×	×
	215-SATASAEGTAIG-226	×	×	

**Table 2 pathogens-10-01250-t002:** Antigenicity evaluation of the predicted epitopes.

Protein	Epitope	Length (mers)	Vaxijen Score
OMP-H	OMP-H_(51–59)_	9	0.524
	OMP-H_(91–106)_	16	0.671
Com-1	Com-1_(26–34)_	9	1.435
	Com-1_(57–76)_	20	0.923
	Com-1_(83–96)_	14	0.139
	Com-1_(191–206)_	16	0.418
OMP-P1	OMP-P1_(43–51)_	9	0.383
	OMP-P1_(98–106)_	9	0.598
	OMP-P1_(197–209)_	13	0.598
	OMP-P1_(215–227)_	13	1.359

**Table 3 pathogens-10-01250-t003:** Summary of the studied population’s clinical and epidemiological data.

	Overall (*n* = 57)	*C. burnetii* Seroreactive (*n* = 26)	*C. burnetii* Non-Reactive (*n* = 31)
Age-Median (IR)	33 (22–49)	33 (21–53)	33 (23–48)
Gender, *n* (%)			
Male	32 (66%)	17 (65%)	15 (48%)
Female	25 (42%)	9 (35%)	16 (52%)
Symptomatology, *n* (%)			
Fever	24 (42%)	10 (38%)	14 (45%)
Nausea	7 (12%)	4 (15%)	3 (10%)
Endocarditis	6 (11%)	4 (15%)	2 (6%)
Hemorrhagic manifestations	1 (2%)	0	1 (3%)
Myalgia	13 (23%)	4 (15%)	9 (29%)
Respiratory manifestations	11 (19%)	3 (12%)	8 (26%)
Prostration	18 (32%)	6 (23%)	12 (39%)

**Table 4 pathogens-10-01250-t004:** Heatmap analysis of *C. burnetii*-seroreactive patients based on reactivity indices against predicted linear B-cell epitopes. Each cell represents the R.I. of one sample against an epitope.

Group	ID	OMP-H_(51–59)_	OMP-H_(91–106)_	Com-1_(57–76)_	Com-1_(191–206)_	OMP-P1_(197–209)_	OMP-P1_(215–227)_
*C. burnetii* **seroreactive patients**	318/12	0.827	0.671	0.813	0.920	0.765	0.912
	487/12	1.006	0.736	1.020	0.856	0.610	0.869
	69/13	0.822	0.736	0.593	0.583	0.869	0.856
	207/17	0.651	0.891	0.890	0.910	0.648	1.060
	464/16	0.791	0.744	0.824	0.774	0.663	0.668
	466/16	1.038	1.279	0.271	1.089	0.471	0.931
	471/16	1.094	1.042	0.946	1.045	1.418	1.259
	104/12	1.430	1.634	1.916	1.887	1.420	1.695
	551/12	1.816	1.601	1.318	0.716	0.849	1.625
	118/13	2.265	1.260	1.408	1.242	1.585	1.224
	463/16	1.100	0.826	1.001	3.540	0.892	1.014
	126/13	1.250	1.475	0.611	0.271	0.689	0.941
	134/13	0.997	0.882	0.837	0.296	0.740	1.757
	130/16	2.075	2.006	1.712	1.180	0.824	2..014
	140/13	1.137	0.193	1.132	1.060	0.915	0.901
	151/13	1.055	0.180	1.137	1.298	0.921	0.877
	204/13	1.250	0.192	1.319	1.197	0.796	1.141
	253/13	0.981	1.744	1.340	1.864	0.993	2.038
	264/13	3.086	4.270	4.916	6.404	0.996	4.994
	304/13	1.369	1.808	1.716	2.371	0.990	1.885
	81/14	0.432	0.498	0.589	0.462	0.814	0.663
	20/16	0.835	0.550	0.664	0.606	0.658	0.818
	480/16	0.412	0.640	0.552	540	0.110	0.628
	490/17	2.072	2.088	2.215	2006	1.338	1.940
	239/18	1.469	1.835	1.948	1676	1.216	1.604
	311/18	2.042	2.497	2.372	2256	1.379	2.175

Each cell represents the R.I. of one sample against an epitope. Each line represents R.I. of one C. burnetii seroreactive patient against each studied peptide (colums). Non-responders (R.I. ≤ 1) are indicated by light blue cells; low responders (1.0 > R.I. ≤ 2.0) are indicated by orange cells, and high responders (R.I. > 2.0) are indicated by dark red cells.

## Data Availability

Publicly available datasets were analyzed in this study. This data can be found here: https://www.uniprot.org/ accessed on 16 August 2021.

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
