# Peer review of "Identification of Immunogenic Linear B-Cell Epitopes in C. burnetii Outer Membrane Proteins Using Immunoinformatics Approaches Reveals Potential Targets of Persistent Infections"

_pathogens, 2021, doi:10.3390/pathogens10101250_

Round 1
Reviewer 1 Report
Manuscript: pathogens-1301939
"Identification of immunogenic linear B-cell epitopes in C. burnetii outer membrane proteins using immunoinformatic approaches reveals potential targets of persistent infections." by da Silva Fontes et al.
The authors performed an in silico analysis to predict linear B-cell epitopes of three C. burnetii outer membrane proteins, 25 OMP-H (CBU_0612), Com-1 (CBU_1910), and OMP-P1 (CBU_0311). The authors identified two linear B-cell epitopes in each protein and confirmed their natural immunogenicity by ELISA. Samples of Q fever patients were screened for the occurrence of the synthetic peptides to validate their relevance. da Silva Fontes et al. report a higher frequency of humoral responses to OMP-H and Com-1 epitopes in patients with endocarditis. The authors propose that their analysis contributes to the further development of serological tools for Q fever diagnosis.
Points of critique:
- The authors describe a strategy to predict, evaluate, and synthesize burnetii-specific linear B-cell epitopes. A flow chart/model for better comprehensibility would be helpful.
- The authors assume that the B-cell epitopes in the burnetii outer membrane proteins reveal potential targets of persistent infections - despite limited numbers of patients with endocarditis (n=6) and the fact that also non-endocarditis cases show a specific reaction to OMP-H. This assumption needs to be discussed more intensively and put into perspective.
- The manuscript leaves in parts a rather sloppy impression. Abbreviations are not introduced constantly, and the phrasing is not designed uniformly. Many sentences are very long and difficult to read. Further correction and spell check is recommended.
Author Response
Dear Reviewer,
Firstly, we thank you for your comments and suggestions. Based on these, we revise the manuscript and believe that now you can accept it.
All responses and modifications are indicated below.
Best regards,
Rodrigo Nunes
The authors describe a strategy to predict, evaluate, and synthesize burnetii-specific linear B-cell epitopes. A flow chart/model for better comprehensibility would be helpful.
Thanks for suggestion. Based on this comment, we insert a new figure in the article, the “Figure 5”. Schematic summary of B-cell epitopes prediction (Please see the attachment.).
The authors assume that the B-cell epitopes in the burnetii outer membrane proteins reveal potential targets of persistent infections - despite limited numbers of patients with endocarditis (n=6) and the fact that also non-endocarditis cases show a specific reaction to OMP-H. This assumption needs to be discussed more intensively and put into perspective.
We are in accordance with this observation. Unfortunately, the lack of similar studies, specially focused in Q fever and its complications, makes difficult a more intense discussion. Despite this, we changed the text in order to highlight the necessity of more studies to verify associations between high antibodies levels against described OMP-H and Com-1 epitopes and Q fever chronic complications. The text was modified in discussion as show bellow:
“Regarding the response against each epitope, we observed that endocarditis seems to be more frequent in HR to OMP-H and Com-1 epitopes (Figure 4), suggesting that an in-tense humoral response to Com-1 and OMP-H epitopes may be associated with Q fever chronic complications. This hypothesis corroborates the potential of recombinant OMP-H as an antigen to serological diagnosis of C. burnetii endocarditis cases [37], however more studies involving a higher number of patients and complications still are necessary to prove this hypothesis.”
The manuscript leaves in parts a rather sloppy impression. Abbreviations are not introduced constantly, and the phrasing is not designed uniformly. Many sentences are very long and difficult to read. Further correction and spell check is recommended.
We apologize for this. The manuscript was revised to rectify these points.

Reviewer 2 Report
1st Revision Pathogens
In the manuscript by da Silva Fontes et al., the authors report bioinformatics-based identification of B-cell epitopes from C. burnetii outer membrane proteins, a class 3 pathogen responsible for the zoonosis Q Fever. The authors confirm the natural immunogenicity of these predicted epitopes which may be valuable tool for Q fever diagnosis. The manuscript is of interest for scientists of the field as Q fever cases remain mainly unreported and or underestimated. However, I have several concerns that must be addressed:
Major comments
The manuscript needs to be check for plagiarism. For example, lines 42-44 are very similar to the sentence found in the review by Burette & Bonazzi, 2020 entitled “From neglected to dissected: How technological advances are leading the way to the study of Coxiella burnetii pathogenesis”.
The authors focus their study on three C. burnetii outer membrane proteins (OMP-P1, OMP-h and Com-1). However, the rationale behind the choice of the studied proteins is not presented here, as it exists additional membrane proteins as shown by Flores-Ramirez et al., 2009. For example, the invasin OmpA is not presented and studied here while it has been identified as an important C. burnetii virulence factor by Martinez et al., 2014; van Schaik et al., 2017.
In the context of the development of efficient serological tests, genetic polymorphisms could be an issue. For the studied OMPs, does B-cell epitope polymorphisms of OMPs have been described among C. burnetii strains ? In case of polymorphisms, what C. burnetii strains are found in Brazil ? The authors must then address if the predicted B-cell epitopes identified in silico in the first part of the manuscript are found in these strains.
Figure 2: The way the figure is presented is not clear and does not contain the appropriate controls. There is no information regarding at what time the samples were collected during the disease course.
Furthermore, and most importantly, the figure does not contain a negative control as samples from patients that have never been exposed to C. burnetii.
Minor comments
Line 41: Remove the additional ‘,’
Line 54: Remove the additional “the”
Line 66: “It’s” should be replaced by “its”
Lines 88, 99 : Remove the uppercase “coxiellosis”
Author Response
Response to reviewer 2
Dear Reviewer,
Firstly, we thank you for your comments and suggestions. Based on these, we revise the manuscript and believe that now you can accept it.
All responses and modifications are listed below.
Best regards,
Rodrigo Nunes
The manuscript needs to be check for plagiarism. For example, lines 42-44 are very similar to the sentence found in the review by Burette & Bonazzi, 2020 entitled “From neglected to dissected: How technological advances are leading the way to the study of Coxiella burnetii pathogenesis”.
Based on your comment, after the manuscript revision, we check it for plagiarism in three online servers (https://app.grammarly.com/), (https://smallseotools.com/pt/plagiarism-checker/) (https://www.plagiarismchecker.co/pt), to certify the non-plagiarism.
The authors focus their study on three C. burnetii outer membrane proteins (OMP-P1, OMP-h and Com-1). However, the rationale behind the choice of the studied proteins is not presented here, as it exists additional membrane proteins as shown by Flores-Ramirez et al., 2009. For example, the invasin OmpA is not presented and studied here while it has been identified as an important C. burnetii virulence factor by Martinez et al., 2014; van Schaik et al., 2017.
Thank you for your comment. To explain the rationale behind the choice of studied proteins, we changed the Introduction and inserted the text in lines 88-97, as show below.
In this context, numerous novel antigens candidates have been proposed to improve the Q fever serodiagnosis [36-40]. Among these antigens, OMP-H (CBU_0612) was described as an immunodominant marker for acute and persistent forms of Q fever [35,41]; Com-1 (CBU_1910) is considered a reliable Q fever serodiagnosis marker [42,43] and OMP-P1 (CBU_0311) is a porin, highly expressed only in the replicative form of bacterium in the cell hosts [44-46]. The identification of B-cell epitopes arises as a promising alternative to improve the specificity of serological tests to C. burnetii, since the use in serodiagnosis of above-mentioned whole antigens may result into a cross reactivity among others phylogenetically related proteobacteria.
In the context of the development of efficient serological tests, genetic polymorphisms could be an issue. For the studied OMPs, does B-cell epitope polymorphisms of OMPs have been described among C. burnetii strains? In case of polymorphisms, what C. burnetii strains are found in Brazil? The authors must then address if the predicted B-cell epitopes identified in silico in the first part of the manuscript are found in these strains.
Regarding this comment, we agree that B-cell epitopes conservation among C. burnetii strains may be a critical aspect to their further use as serological markers. Unfortunately, the global distribution of C. burnetii strains are still poorly understood. Taking it into consideration, it is important to remind that there is a lack of information about C. burnetii genotypes in the world. The full amino acid sequences of proteins OMP-H and Com-1 are only described to strains RSA 493 / Nine Mile phase I (OMP-H: CBU_1910; Com-1:CBU_1910) and Dugway 5J108-111 (OMP-H: CBUD_0624; COM-1: CBUD_0209), while the full sequence of OMP-P1 is reported only to strain RSA 493 / Nine Mile phase I (CBU_0311). In order to evaluate the question raised by the referee, we analyzed the protein sequence conservation among a few available strains, and we observed that there that there is no significant aminoacid alterations in the regions comprised by the predicted B-cell epitopes. In this sense, we can assume that the regions predicted as B-cell epitopes probably are the same between different C. burnetii strains. In addition to this data, Mioni and collaborators (2020) published the still unique study that described the genotypes of C. burnetii circulating in Brazil and Argentina, reporting the presence of novel multispacer sequence typing (MST) genotypes of C. burnetii in two clusters and describing that Brazilian strains from goats (CbG_SVB22), cattle (CbB_F2), and ovine (CbO_sn2), presented the same MST loci combination, seem to be the same MST clone of C. burnetii.
Figure 2: The way the figure is presented is not clear and does not contain the appropriate controls. There is no information regarding at what time the samples were collected during the disease course.
Furthermore, and most importantly, the figure does not contain a negative control as samples from patients that have never been exposed to C. burnetii.
Regarding the negative control composed by “samples from patients who never have been exposed to C. burnetii”, the lack of national surveillance to C. burnetii hamper this strategy. By this way and considering that serological diagnosis indicates exposition/contact to pathogen, not necessarily an active infection, we believe that samples from patients who were not reactive to C. burnetii in immunofluorescence assay is a reliable negative control group. Besides, about “what time the samples were collected during the disease course”, unfortunately, we don’t have this information, because in Brazil only rickettsiosis suspect cases are investigated to Q fever. By this way, we can not specify the time during the disease course in that the collection was done. However, considering that the main aim of our study was to evaluate the immunogenicity of predicted B-cell epitopes in naturally exposed population to C. burnetii, we believe that this strategy was enough to prove our point. In other hand, to improve the presentation of Figure 2, we modified its fonts, size, and legends.
Minor comments
Line 41: Remove the additional ‘,’
Line 54: Remove the additional “the”
Line 66: “It’s” should be replaced by “its”
Lines 88, 99 : Remove the uppercase “coxiellosis”
We apologize for this. All minor comments were corrected in the manuscript.
Round 2
Reviewer 2 Report
In the rebuttal letter, the authors clair that the revised manuscript has been carrefully check for plagiarism. I agree for this point.
However, some of my concerns still persist. Regarding the rationale behind the choice of the studied proteins, the authors do not answer the request on this aspect of the manuscript. There is no explanation on why they excluded the study of the other outer membrane protein and especially the invasion OmpA which is a key C. burnetii virulence factor. Comprehensive analysis of putative immunogenic OMPs would strenghthen the rationale of the paper.
Regarding genetic polymorphism of OMPs, the authors analyzed the protein sequence conservation among strains, however this analysis is not shown and should be to strenghthen the revised manuscript. For example, if the sequence of OMP-P1 is reported only in Nine Mile I strain, how the authors can assume the fact that there is no significant aminoacid alterations in the region comprising the predicted epitopes ?
Author Response
Dear Reviewer 2,
Firstly, we appreciate the valuable comments and suggestions received from the reviewers that helped to substantially improve our manuscript. Below you will find an itemized list of the changes made in the text of revised version of to address these lasts specific concerns.
All responses and modifications are listed below.
Best regards,
Rodrigo Nunes
In the rebuttal letter, the authors clair that the revised manuscript has been carrefully check for plagiarism. I agree for this point.
Thank You.
However, some of my concerns still persist. Regarding the rationale behind the choice of the studied proteins, the authors do not answer the request on this aspect of the manuscript. There is no explanation on why they excluded the study of the other outer membrane protein and especially the invasion OmpA which is a key C. burnetii virulence factor. Comprehensive analysis of putative immunogenic OMPs would strength then the rationale of the paper.
We understand your comment and totally agree on the importance of studying OmpA also as other immunogenic proteins. However, this manuscript is a component of a better study where we identified diagnostic targets in several proteins like OmpA, Cb-Mip, YbgF among others. Thus, all data concerning OmpA are being explored in another work, which aimed vaccine development through the epitope identification, assembly of monoclonal antibodies against each epitope and evaluation of their potential to recognize and inhibit bacterial interactions. Therefore, considering the scope of this present study, which aimed to verify the natural immunogenicity of B-cell linear epitopes to the serodiagnosis and to not the vaccine development, the association between the studied proteins and virulence wasn't considered as a key factor to their selection.
On other hand, considering your valuable comment we modify the discussion and insert the text as shown below:
“These data reinforce the need of to spot B-cell linear epitopes in other promising serological markers, like as OmpA [89], C. burnetii macrophage infectivity potentiator protein (Cb-Mip) [90] and YbgF [85], to improve the development of novel tools to C. burnetii diagnostic, interventonal therapies and vaccine development.”
Regarding genetic polymorphism of OMPs, the authors analyzed the protein sequence conservation among strains, however this analysis is not shown and should be to strength then the revised manuscript. For example, if the sequence of OMP-P1 is reported only in Nine Mile I strain, how the authors can assume the fact that there is no significant amino acid alterations in the region comprising the predicted epitopes?
Thank you for your comment. We understand and agree with your point of view, and however, due to the scarcity of studies focused on polymorphism of Omps, we also believe that this analysis is going to be little informative, and additionally, could create an unsupported epitope conservation among C. burnetii strains. To clarify this point, and in accordance with your opinion, we revise the discussion by inserting the text below:
"Besides, the specificities of predicted epitopes were verified by the comparison with proteobacteria proteins. Remarkably, despite all predicted epitopes were conserved among C. burnetii strains described in the Uniprot database (data not showed), the scarcity of studies concerning OMPs polymorphism hamper conclusions about the real conservation of the identified epitopes in C. burnetii strains."
Round 3
Reviewer 2 Report
In the rebuttal letter, responses are complete and additional modifications strengthen the revised form of the manuscript. I have no additional concern regarding this paper and I’m looking forward to hear about OmpA work.